# Trauma Quality Improvement Program: A Retrospective Analysis from A Middle Eastern National Trauma Center

**DOI:** 10.3390/healthcare11212865

**Published:** 2023-10-31

**Authors:** Hassan Al-Thani, Ayman El-Menyar, Naushad Ahmad Khan, Rafael Consunji, Gladys Mendez, Tarik S. Abulkhair, Monira Mollazehi, Ruben Peralta, Husham Abdelrahman, Talat Chughtai, Sandro Rizoli

**Affiliations:** 1Department of Surgery, Trauma Surgery, Hamad Medical Corporation, Doha 3050, Qatar; althanih@hotmail.com (H.A.-T.); rconsunji@hamad.qa (R.C.); gmendez@hamad.qa (G.M.); tabuelkkeer@hamad.qa (T.S.A.); mollazehi@hamad.qa (M.M.); rperaltamd@gmail.com (R.P.); hushamco@hotmail.com (H.A.); tchughtai@hamad.qa (T.C.); srizoli@hamad.qa (S.R.); 2Clinical Research, Trauma & Vascular Surgery Section, Hamad Medical Corporation, Doha 3050, Qatar; nkhan13@hamad.qa; 3Department of Clinical Medicine, Weill Cornell Medicine, Doha 3050, Qatar; 4Department of Surgery, Universidad Nacional Pedro Henriquez Urena, Santo Domingo 10100, Dominican Republic

**Keywords:** ACS-TQIP, benchmarking, quality improvement, trauma center, injury, trauma system

## Abstract

Background: The Trauma Quality Improvement Program (American College of Surgery (ACS-TQIP)) uses the existing infrastructure of the Committee on Trauma programs and provides feedback to participating hospitals on risk-adjusted outcomes. This study aimed to analyze and compare the performance of the Level I Hamad Trauma Centre (HTC) with other TQIP participating centers by comparing TQIP aggregate database reports. The primary goal was to pinpoint the variations in adult trauma outcomes and quality measures, identify areas that need improvement, and leverage existing resources to facilitate quality improvement. Methods: A retrospective analysis was performed for the TQIP data from April 2019–March 2020 to April 2020–March 2021. We used the TQIP methodology, inclusion and exclusion criteria, and outcomes. Results: There were 915 patients from Fall 2020 and 884 patients from Fall 2021 that qualified for the TQIP database. The HTC patients’ demographics differed from the TQIP’s aggregate data; they were younger, more predominantly male, and had significantly different mechanisms of injury (MOI) with more traffic-related blunt trauma. Penetrating injuries were more severe in the other centers. During the TQIP Fall 2020 report, the HTC was a low outlier (good performer) in one cohort (all patients) and an average performer in the remaining cohorts. However, during Fall 2021, the HTC showed an improvement and was a low outlier in two cohorts (all patients and severe TBI patients). Overall, the HTC remained an average performer during the report cycles. Conclusions: There was an improvement over time in the risk-adjusted mortality, which reflects the continuous and demanding effort put together by the trauma team. The ACS-TQIP for the external benchmarking of quality improvement could be a contributor to better monitored patient care. Evaluating the TQIP data with emphases on appropriate methodologies, quality measurements, corrective measures, and accurate reporting is warranted.

## 1. Introduction

Traumatic injury remains the leading and most substantial cause of preventable morbidity and mortality worldwide, with a global economic burden of hundreds of billions of dollars [1,2]. Trauma is the fourth leading cause of death in Qatar (7.27 deaths per 100,000 population) [3]. Given these facts, the goal is to prevent all injuries from occurring, as well as provide high-quality trauma patient care and pursue opportunities for improvement (OFI). To this end, it is pertinent to identify deficiencies at all levels of the trauma care system to prevent mortalities and reduce short- and long-term disabilities.

Over the past several years, substantial investments worldwide have been made to set up large clinical registries resembling the National Trauma Data Bank (NTDB) in the USA. The NTDB data have been increasingly used in trauma research to provide outcome benchmarks for hospital quality improvement [4,5]. Despite progress being made, trauma remains an important and underestimated cause of morbidity and mortality. One of the hallmarks of organized trauma care systems is the commitment to continually evaluate outcomes and search for opportunities to improve care [6,7]. However, some challenges include identifying opportunities and creating a benchmark against which to measure existing trauma care performance. To address this, a recognized program, the Trauma Quality Improvement Program (TQIP), was designed by the American College of Surgeons (ACS-COT) Committee on Trauma, pilot tested in June 2008, and opened for formal enrolment in 2010 to evaluate the quality of trauma care provided, identify areas in need of improvement, and reduce trauma mortality based on predictive modeling techniques [8,9,10]. The TQIP model is a validated, risk-adjusted, outcomes-based benchmarking model to predict mortality for all injured adult patients by using data extracted from standard medical record information from a large clinical data repository. The TQIP achieves its objectives by collecting data from level I and II trauma centers, providing feedback about its performance, and identifying institutional characteristics that the trauma center staff can modify to improve patient outcomes by providing accurate national comparisons [11]. This shared collaboration can yield best practices by providing a scientific basis for the focused implementation of quality improvement programs for the enrolled centers [10] and by laying out the resources and processes required to provide high-quality care to the injured [12]. A Level 1 trauma center is a specialized medical facility dedicated to comprehensive trauma care, covering prevention, treatment, and rehabilitation services. It operates 24/7 and has a team of specialized surgeons, including orthopedics and neurosurgeons. This center offers a prompt and wide range of specialized services such as pre-hospital care, emergency medicine, orthopedics, neurosurgery, radiology, critical care, and others, ensuring swift and diverse care for trauma cases. Quality assessment, research initiatives, and educational programs contribute to the ongoing enhancement of trauma care and meet a minimum requirement regarding the annual volume of severely injured patients. On the other hand, a Level 2 trauma center collaborates with Level 1 centers to provide continuous trauma care and has access to essential specialties, personnel, and equipment. While they can handle most types of injuries, Level 2 centers are not obligated to maintain research and educational programs or meet the same research expectations as Level 1 centers.

One of the TQIP’s most important features is that it allows for validation and risk-adjusted analyses, which are crucial for assessing trauma outcomes [13]. However, despite having a well-accepted standard of care and an excellent ability to predict outcomes in adult patients, there still appears to be notable variability in the care and outcomes between trauma centers [12,14]. The current study sought to analyze and compare the clinical presentation and outcomes of trauma patients at the HTC, which is a contributing institution to the TQIP, with data derived from the Qatar National Trauma Registry (QNTR). This was compared with the aggregate data from the other trauma centers that participated in and provided data to the TQIP. Furthermore, we sought to identify variations in adult trauma outcomes and quality measures, highlight areas requiring improvements, and utilize available resources to facilitate quality improvement. We hypothesized that the implementation and quarterly auditing of the Standard Operating Procedures (SOPs) of the TQIP guidelines would result in an overall survival benefit and an improvement in clinical outcomes.

## 2. Materials and Methods

### 2.1. Data Source and Study Design

This is a retrospective cohort study of all of the trauma patients who were admitted to the tertiary Level 1 Hamad Trauma Centre (HTC) at the Hamad General Hospital (HGH), whose data were encoded and captured in the QNTR and accepted for inclusion in the TQIP database, i.e., TQIP-eligible. The HTC is the only Level 1 national trauma referral center in Qatar; it provides trauma care for all of its residents and citizens (2.8 million inhabitants) free of charge. The analyzed and compared data were extracted from two annual TQIP reporting cycles, Fall 2020 (April 2019–March 2020) to Fall 2021 (April 2020–March 2021), and TQIP benchmark reports submitted to the TQIP database.

### 2.2. Existing Model (TQIP)

The TQIP model is a risk-adjusted, outcomes-based measure created to improve the quality of trauma care. Its purpose is to predict mortality in injured adult (≥16 years) patients as a function of the following covariates (sorted from most to least important): the initial motor Glasgow Coma Scale (GCS) score in the emergency department (ED), initial systolic blood pressure (SBP) in the ED, Injury Severity Score (ISS), age, initial pulse rate in the ED, mechanism of injury, head injury severity, abdominal injury severity, and patient transfer status. The TQIP model generates an impressive c-statistic of 0.901 across the wide spectrum of TQIP-eligible patients, from 16 to greater than 100 years of chronological age, but, like any model of physiological phenomena, it is unlikely to perform equally well across the multiple strata of age. The TQIP tracks all patients’ mortalities and complications. The TQIP reported on the following eight patient cohorts in the Fall 2020 and Fall 2021 TQIP benchmark report: (1) all patients, (2) blunt multisystem injuries, (3) patients with penetrating injuries, (4) patients with shock, (5) patients with Severe Traumatic Brain Injury (TBI), (6) elderly patients, (7) elderly patients with blunt multisystem injuries, and (8) elderly patients with isolated hip fractures (IHFs). Statistical models were employed to generate risk-adjusted estimates for the complications and outcomes for each of these cohorts. A summary of the TQIP cohorts at the HTC and the ACS TQIP centers is outlined in Figure 1.

The Hamad Trauma Centre (HTC): The HTC is Qatar’s only national tertiary trauma care center. It is located at Hamad General Hospital (HGH), the hub of a not-for-profit governmental healthcare system. The trauma system in Qatar began to progress in late 2007, with the formation of the trauma surgery unit at HGH as part of an enhanced emergency care collaboration with the University of Pittsburgh. It includes all components of the trauma system, starting from pre-hospital ambulance service, in-hospital management, and rehabilitation. These components have evolved and have undergone rapid change over the years by accomplishing numerous major developmental milestones, including the creation and development of the trauma resuscitation unit, the QNTR, the Trauma Hamad Injury Prevention Program (HIPP), the Trauma and Critical Care Fellowship Program (TCCFP), and the Clinical Research Unit [15,16].

Furthermore, the Trauma Section has evolved into an internationally recognized Centre of Excellence in Trauma Care, the Hamad Level 1 Trauma Centre. In 2013, the HTC became a participant in the ACS-COT TQIP. In 2014, the Qatar Trauma System was the first in the Middle East to be awarded the Trauma Distinction Award by Accreditation Canada International for providing high-quality trauma treatment to critically injured patients. The HGH Trauma Registry is a part of the QNTR that has regular auditing and regular departmental checking for accuracy to minimize the rate of missing data. The trauma registry statistics for the HGH are gathered countrywide. In 2017, the QNTR was formally inaugurated, contributing to the American College of Surgeons (ACS; Chicago, IL, USA) Committee on Trauma’s NTDB, the world’s largest international trauma registry database. The HGH Trauma Registry has international and local validations. Quarterly reporting to the NTDB and the ACS-TQIP is part of the international recognition and validation [16]. The International Classification of Diseases, 10th Revision, the Clinical Modification (ICD-10-CM) code, and the NTDS dictionary are used to update data definitions.

The HTC comprises several key units for patient care: a Trauma Resuscitation Unit (TRU), a Trauma Intensive Care Unit (TICU), a Trauma In-Patient Ward (TSU), and a Trauma Outpatient Clinic. The TRU contains five trauma bays, advanced technology, and a skilled team of trauma surgeons and nurses. It is directly accessible to injured patients via ambulance services, including air and ground transport. The TICU, with 19 beds, provides critical care, following evidence-based guidelines for over 600 trauma cases yearly. A 7-bed trauma step-down unit facilitates the transition from the TICU to in-patient care. The twenty-bed trauma surgery unit handles patient reevaluation, treatment, and discharge. Moreover, post-hospital care involves rehabilitation services available through the National Qatar Rehabilitation Institute and the Trauma Outpatient and Psychiatric Clinic at HMC 1 [15].

### 2.3. Inclusion Criteria

The inclusion criteria for this study included individuals aged 16 years or older who had experienced either blunt or penetrating trauma and had at least one valid abbreviated injury score (AIS) of 05/08, with a severity falling within the range of 3–6 in AIS chapters 1–8, or an equivalent AIS 2015 injury. The primary mechanism of injury needed to be either penetrating or blunt, with “blunt” defined as injuries associated with specific E-code categories, including falls, machinery accidents, motor vehicle traffic incidents, pedestrian accidents, cycling accidents, and being struck by or against an object. “Penetrating” injury was defined as an injury where the primary E-code is mapped to the cut/pierce area and firearm. Additionally, eligible patients must have had at least one AIS score of 3 or higher and blunt and penetrating injuries (as defined in Table 1), and information on both Emergency Department (ED) discharge disposition and hospital discharge disposition needed to be available.

Table 1 summarizes the definitions used in the TQIP.

### 2.4. Exclusion Criteria

Patients with a pre-existing advanced directive specifying the withholding of life-sustaining interventions were excluded. Also, patients who were discharged more than 30 days after their final admission date were not eligible. Additionally, patients who exhibited no signs of life during the initial evaluation, as determined by criteria such as an emergency department (ED) systolic blood pressure of 0, a pulse rate of 0, and a Glasgow Coma Scale (GCS) motor score of 1, and patients with severe burns were also not included.

### 2.5. Variable Selection/Primary Outcome

Collected data points that were included: Demographic information (age, gender, and race/ethnicity); mechanisms of injury (blunt and penetrating); mode of injury (fall, pedestrian, struck by/against, firearm, cut/piece, motor vehicle trauma (MVT), motorcyclist, MVT occupant, and other); pre-existing comorbidities (congestive heart failure (CHF), myocardial infraction (MI), cerebrovascular accident (CVA), hypertension (HT), chronic renal failure (CRF), diabetes mellitus (DM), chronic obstructive pulmonary disease (COPD), cirrhosis, acute respiratory distress syndrome (ARDS), dementia/mental disorders, disseminated cancer, smoking, and alcohol use disorder); and in-hospital complications (acute kidney injury (AKI), pressure ulcer, deep vein thrombosis (DVT), pulmonary embolism (PE), catheter-associated urinary tract infection (CAUTI), ventilator associated-pneumonia (VAP), severe sepsis; hemorrhagic shock; venous thromboembolism (VTE) parameters; Glasgow Coma Score (GCS), Injury Severity Score (ISS); blood transfusion; use of intensive care unit (ICU), ventilatory days, and hospital length of stay (HLOS) in-hospital mortality).

Statistical analysis: Data were reported as proportion, mean (±standard deviation), median, range, or IQR. Patients were categorized into two groups (all TQIP centers vs. HTC TIQP cohort). Results were compared with data published by the TQIP. The Chi-squared test was used to compare proportions between the groups at a significant level (*p*-value < 0.05). Means of quantitative variables were compared using Student’s t-test. Medians were compared using non-parametric tests. To assess the performance of our institution, we reported the TQIP-validated risk-adjusted mortality data to compare the injury severity and outcome variables between our center and the TQIP aggregate database. Data analysis was conducted using the online EpiInfo^TM^ software (Version 3.5.3; Centers for Disease Control and Prevention (CDC) in Atlanta, Georgia (US)).

## 3. Results

From the QNTR, there were 1605 trauma activations recorded during the Fall cycle of 2020 and 1696 trauma activations recorded during the Fall cycle of 2021.

The patient characteristics of the HTC cohort and TQIP aggregate database, including age, gender, injury type, and mechanism of injury, are listed in Table 2. Significant differences were observed in age at the HTC versus the TQIP aggregate; the patients were younger, with male preponderance, compared to the TQIP aggregate data during the two Fall cycles. The HTC recorded a significantly lower proportion of elderly patients (5.0% vs. 37.7 (Fall 2020); 5.7% vs. 35.2% (Fall 2021) *p* < 0.001) versus the TQIP aggregate data. Concerning the mechanism of injury, the proportion of penetrating trauma was significantly lower at the HTC versus the TQIP aggregate data (*p* < 0.001), and the proportion of blunt trauma was significantly higher (*p* = 0.01).

The HTC reported higher proportions of MVT (occupant and other), MVT involving a motorcycle or pedestrian struck by/against an object (*p* < 0.001), and fewer falls (*p* < 0.001) as the mechanism of injury when compared to the TQIP aggregate data. The other MOIs were not significantly different. Among the comorbidities, the most frequently identified comorbidity in the HTC cohort and the TQIP aggregate database were hypertension (10.6%), diabetes mellitus (10.8%), and mental/emotional disorders. A comparative analysis of trauma patients according to their pre-existing comorbidities demonstrated overall significantly lower percentages of hypertension, diabetes mellitus, heart failure, myocardial infarction, cerebrovascular accidents, and cirrhosis (*p* < 0.001) patients at the HTC compared to the TQIP cohorts. All other comorbidities (ARDS, cancer, and AKI) were not significantly different (Table 3).

The HTC received a significantly higher proportion of severe TBI and shock patients when compared to the TQIP database (*p* < 0.001). Higher proportions of VAP, CAUTIs, and CLABSIs (*p* < 0.001) were documented, which also differed significantly from the TQIP aggregate data. Our institution’s report revealed a lower percentage of DVTs and severe cases of sepsis) when compared to the TQIP aggregate database (*p* < 0.001) (Table 4).

The TQIP specifically targeted the management of hemorrhagic shock and VTE prophylaxis as a quality improvement focus. Among trauma patients, major hemorrhages are responsible for 30 to 40% of mortality, with up to half of the patients dying before arriving at the hospital. We compared the hemorrhagic shock management of patients at the HTC with the TQIP aggregate data. During the study period (April 2020–March 2021), the HTC received a significantly higher proportion of hemorrhagic shock patients compared to the TQIP database (*p* < 0.001). A higher proportion received Packed Red Blood Cell (PRBC) transfusion and required the angiographic management of a hemorrhage (*p* < 0.001) at the HTC, which also differed significantly from the TQIP aggregate data. The patients at our center received lower proportions of platelets and plasma transfusions within 24 h than those in the TQIP database (Table 5).

The HTC had a significantly higher proportion of patients who received pharmacologic VTE prophylaxis than the TQIP aggregate cohort. The VTE prophylaxis compliance data for the cohort in this study demonstrated a greater use of low-molecular-weight heparin as the drug type compared to the TQIP aggregate database. Comparable and good compliance with the timing of administration for the first dose of pharmacologic VTE prophylaxis was also observed with a median time of 2 days (within 48 h after admission) and hospital stay longer than two days for the HTC cohort.

Table 6 outlines the characteristics and comparison of injury severities and different outcome variables between the HTC and the TQIP aggregate database. The HTC showed a significantly higher proportion of severely injured patients with a GCS score equal to 8 or less. The median ISS of our cohort was found to be like the other TQIP participating centers. The patients at our institution had a longer median length of hospital stay, ventilatory days, and ICU length of stay compared to the TQIP aggregate database. The difference between the median length of hospital stays, ventilator days, and ICU length of stay was found to be statistically significant among the patients at the HTC compared to the other TQIP participating centers. A significantly lower percentage of patients died within 72 h and after 30 days of hospital admission, respectively, at the HTC, compared to the TQIP aggregate data. Also, the HTC recorded a significantly longer median time to death after hospital admission than the TQIP participating centers.

Table 7 outlines the risk-adjusted mortality. The HTC was a low outlier, or good performer, in the “all patient” and “severe TBI” cohorts, but it was an average performer in the remaining cohorts. The risk-adjusted observed/expected (O/E) ratio for aggregate mortality for all patients at the HTC for Fall 2020 was 0.58 (95% CI, 0.40–0.84) and 0.47 (95% CI, 0.32–0.67) for Fall 2021. The risk-adjusted O/E ratio for the severe TBI cohort for Fall 2020 was 0.64 (95% CI, 0.40–1.01), and it improved to 0.42 (95% CI, 0.26–0.68) for Fall 2021. Although the risk-adjusted O/E ratio for major hospital events, including mortality, showed a slight improvement during Fall 2021 in all patients, severe TBI, and shock cohorts, it did not reach statistical significance based on an odds ratio (OR) and the confidence interval values (Figure 2).

In the modified box plot, a diamond represents the hospital’s estimated outcome, and a line extends to indicate the length of the associated confidence interval. Additionally, this modified box plot highlights the hospital-level median estimate. The green box corresponds to the 1st decile, while the pink box represents the 10th decile. Between these two boxes, a rectangular gray box displays data spanning from the 10th to the 90th percentile. Moreover, a smaller gray box within the rectangular one signifies the Interquartile range (IQR). The upper 10th decile is depicted in a pink box, encompassing the entire TQIP sample. If the odds ratio (diamond) falls within the green box (1st decile), it indicates a low outlier or a well-performing hospital. Conversely, when the diamond falls in the upper first decile (red box), it suggests low performance or a high outlier. If the diamond does not fall within the green or red boxes, it represents average performance and is displayed as a black diamond.

## 4. Discussion

This study shows that at the HTC, the TQIP data regularly appraises the performance and identifies areas for improvement. The TQIP helps accomplish this by using a risk-adjusted model for mortality prediction using real-time data [13], which allows for considering variations in the case mix for improving the quality of care provided at trauma centers through inter-institutional comparisons [14,15]. The TQIP also seeks to understand the reasons for heterogeneity in trauma management, learn from high-performing hospitals, and give performance feedback to participating trauma centers that would enhance outcomes among trauma patients. Prior publications have documented the inception, feasibility, and methodology used for the risk adjustment and benchmarking of the TQIP [10,11].

Several published studies extol the benefits of a national registry for quality improvement [4,8,10,17,18,19,20]. At our institution, among all admissions, only 42.8% and 44.5% of admissions qualified for the Fall 2020 and 2021 TQIP reports, respectively, and the rest of the patients were excluded from analysis because of the limitations of the TQIP inclusion criteria and the breadth of the exclusion criteria. The patient demographics at our institution differed significantly from those of the TQIP aggregate population. Our population was younger, more predominantly male, and sustained a greater proportion of blunt trauma due to traffic-related injuries.

The QNTR started contributing to the ACS-COT’s TQIP in 2013; this report is the first collaborative initial evaluation and comparison of the clinical outcomes of the national TQIP database (derived from the QNTR) with the other TQIP centers. To the best of our knowledge, our institution is the only active ACS-COT TQIP collaborative center focusing on trauma quality improvement initiatives in the Eastern Mediterranean region. The HTC, at its core, is striving for quality improvement by participating and submitting data to the TQIP and ACS-COT, thereby serving as an example of a progressive and growing healthcare system. Various system-wide protocol modifications have been introduced at our institution during the past decade, including changes to the trauma triage criteria, the implementation of massive transfusion protocols, and the adoption of damage control surgery and damage control resuscitation principles with an expansion in the utility of point-of-care testing and imaging. Quality improvement activities at the HTC take a regional collaborative approach based on fundamental yet important concepts. Performance improvement due to the implementation of quality improvement programs at the local level often involves unique solutions tailored to the specific setting of each hospital. Sharing hospital-centered issues in a collaborative network such as the TQIP can provide access to data and, subsequently, a comparison and flexible incorporation of data elements in a meaningful way [21].

Our findings suggest that the risk-adjusted mortality estimates provided by the TQIP mortality prediction model may not be applicable for all patient populations as numerous characteristics that were shown to be important at the HTC were not significant in the TQIP mortality prediction model, such as the mechanism of injury, shock, and severe TBI. Therefore, we should learn more about the variables that were incorporated into the risk-adjustment models used for mortality prediction in the HTC.

A pilot study from the United States compared the institutional characteristics from the TQIP aggregate data and discovered that several variables that were reported as significant in the TQIP model were not found to be predictors of mortality (age, initial pulse rate, in ED, mechanism of injury, GCS: 2–5; SBP > 90 mm Hg). They concluded that, despite the need for the external benchmarking of trauma center performance using mortality prediction models, the TQIP methodology might not apply to all centers, and variability may exist [18]. However, despite these advancements, the explanatory factors remain ambiguous.

The TQIP was created to provide data to individual institutions on their risk-adjusted outcomes in the form of O/E ratios, which can reveal areas where a hospital has a high, medium, or lower performance compared to its peers [22]. Compared to the TQIP aggregate statistics, the HTC had a larger number of patients who received pharmacologic VTE prophylaxis. Its compliance statistics for the patients at our center showed a higher utilization of low-molecular-weight heparin. Also, the HTC admits a greater number of severe TBI patients than the TQIP aggregate centers. The frequency of VTE in trauma patients is the highest in the first few days after admission. TBI adds to the risk [23], especially when the risk of bleeding prevents a prompt and early delivery of VTE prophylaxis. There is mounting evidence in the literature to suggest starting thromboprophylaxis as soon as feasible after a TBI (within 24–72 h after admission) while considering the stabilization of intracranial/extracranial bleeding and in conjunction with neurosurgeon consultation [23,24,25,26].

We observed comparable and good compliance with the timing of administration for the first dose of VTE prophylaxis with a median time of 2 days for our cohort. The HTC was a low outlier (good performer) in the all-patient cohort and an average performer in the remaining cohorts, according to the TQIP Fall 2020 report. Nonetheless, the TQIP Fall 2021 report demonstrated that our HTC has improved, being a low outlier (good performer) in the “all patient” and “severe TBI” cohorts but remaining as an average outlier in all other cohorts. Also, we identified cohorts with relatively high outlier statuses (i.e., shock, elderly, elderly blunt multisystem, and isolated hip fracture) at our institution in the TQIP data for the two consecutive Fall cycles.

Comparing the HTC set to the TQIP aggregate data enabled us to evaluate our performance and gain insight into the compliance and utilization of chemoprophylaxis for VTE and TBI at our institution. The HTC’s next steps should be to re-evaluate the problem, take corrective action, and develop institution-specific best practice guidelines in conjunction with existing prevention strategies.

Furthermore, efforts should continue to decrease ventilatory days, the length of ICU stay, and overall hospital stay at our institution by providing optimal care to our patients. These goals may be followed in the next TQIP report to see if the implemented action items impacted the outcomes or otherwise to refocus the strategy. At our center, reporting to the TQIP has a role in reducing the critical consequences (VTE and severe TBI) and improving compliance in trauma patients. According to our findings, participating in a TQIP program and benchmarking clinical results for a cohort of trauma patients enhances clinical outcomes.

However, mentioning some challenges while using the TQIP database for performance benchmarking is pertinent. The ACS strictly monitors the TQIP data quality, although employing registry data poses specific issues [17,27]. For instance, some metrics are added, while others are removed for some reasons, including the problem’s scope. In this cost-conscious healthcare environment, updating and compliance with these resources can be difficult for trauma program administrators. Training is necessary to acquaint trauma registry staff with new features, which can be time- and resource-consuming. Finally, program managers must constantly review data entry for errors that might skew the data, interpret the results, and affect the outcomes.

**Action plan**: The TQIP reports help healthcare providers to understand areas where they excel and areas where they fail (outliers), and these statistically adjusted observations are the scientific basis for a change in clinical practice. The failure or problem areas need special attention, correction action plans, and individualized solutions. The subsequent reports would reflect the impact of the applied changes. We identified areas where the center outlies in comparison to others. These areas were prioritized and scrutinized to rule out confounding issues like more screening or coding errors. The data were shared with all involved parties, including hospital administration, under the lead of the performance improvement office, to create an awareness and sense of urgency to drive improvements and changes. A multidisciplinary committee or task group is concerned with each outlier complication list. The members include quality officers, clinicians, consulting service-related nominees, and all related auxiliary services. The tasks include creating or revising evidence-based clinical care guidelines and protocols, definition review, providing education, and identifying high-risk groups of patients and related outcomes. The agreed-upon changes or new bundle of care are then scrutinized, implemented, and reinforced, and the results are followed by compliance data, incidence reports, and registry data, while the impact will be reflected on the subsequent TQIP reports for success and sustainability.

**Limitations**: This study may be subject to several limitations that should be acknowledged. Our study is a retrospective analysis of an extensive database; therefore, a reporting bias may be present. Secondly, The QNTR is a clinical database that may contain missing and minimal information on the process of care. However, for risk-adjustment variables, the amount of missing data was increased to 4.5% in the HTC cohort, whereas it was increased to 7.6% in the aggregate TQIP data. The sample size from the HTC was another limitation in identifying certain outliers in the studied cohort. The TQIP estimations’ precision relies on the sample size, which might be small for some hospitals when examining highly chosen groups (e.g., penetrating injury). This lack of precision may imply that a particular hospital is delivering care at par with other trauma centers; however, estimates with better precision may indicate otherwise. Therefore, the center-based sample size may restrict the utilization of TQIP data to answer specific quality improvement questions. Finally, we did not present the comparison of the quality improvements before and after the inclusion of our institution in the ACS-TQIP database, which would have given a true reflection of our center’s performance concerning the outcome measures. The TQIP reports are released biannually, and it may take several cycles to notice improvements in the TQIP figures after corrective actions are undertaken.

## 5. Conclusions

Applying the appropriate resources and expertise as a part of a continued process of quality improvement programs to achieve quality-driven care could positively improve outcomes. Examining the institution’s data and identifying trends that may be addressed by implementing evidence-based practice changes is critical. The improvements noted in the Fall 2021 vs. Fall 2020 TQIP reports in the risk-adjusted mortality and major hospital events, including mortality, reflected the continuous and arduous effort put together by the trauma program at the HTC. Analyzing the TQIP database reports with a focus on quality measures and accurate reporting will reveal that the TQIP data are beneficial for improving patient care after implementing corrective measures.

## Figures and Tables

**Figure 1 healthcare-11-02865-f001:**
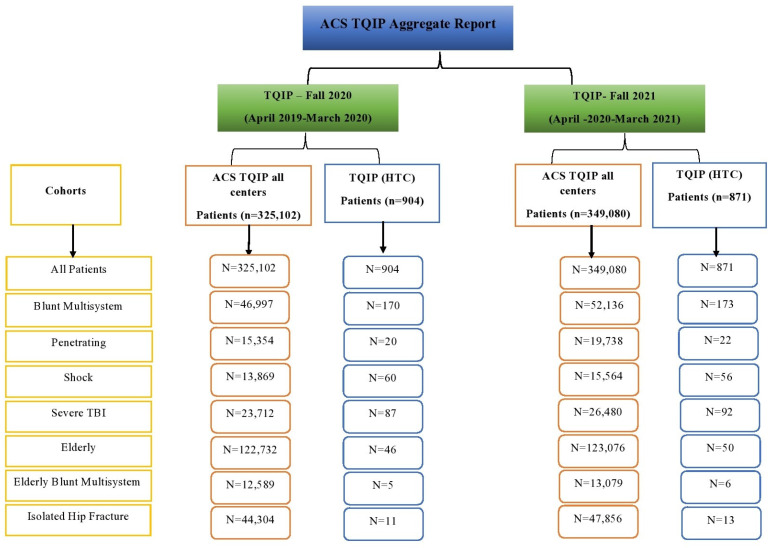
TQIP data for the HTC and all other TQIP participating centers during the two Fall cycles.

**Figure 2 healthcare-11-02865-f002:**
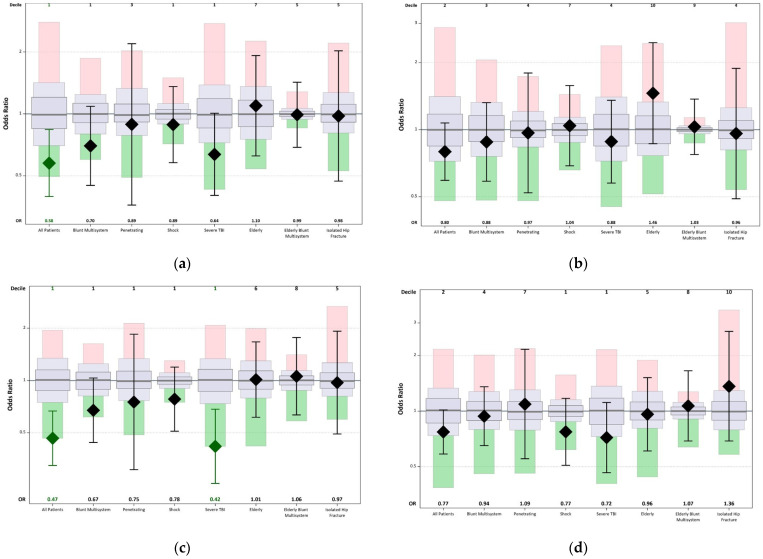
(**a**) TQIP risk-adjusted mortality modified boxplot for Fall 2020. The modified boxplot displays that our institution was a low outlier (good performance) in the all-patient cohort and an average performer in the remaining cohorts (blunt multisystem, penetrating cohorts, shock elderly, elderly blunt multisystem, and isolated hip fracture cohorts). (**b**) TQIP risk-adjusted major hospital events, including mortality-modified boxplot. The modified boxplot displays that the HTC has an average performance in all cohorts. (**c**) TQIP risk-adjusted mortality modified boxplot for Fall 2021. Modified boxplot displays that our institution was a low outlier (good performance) in all patients and severe TBI cohorts, and it was an average performer in the remaining cohorts (blunt multisystem, penetrating cohorts, shock elderly, elderly blunt multisystem, and isolated hip fracture cohorts). (**d**) TQIP risk-adjusted major hospital events, including mortality-modified boxplot. The modified boxplot displays that the HTC has an average performance in all cohorts. In the boxplot, the estimate of the outcome for the HTC is shown as a diamond, and the line from the diamond extends the length of the associated confidence interval. The median hospital-level estimates, as well as the 10th, 25th, 75th, and 90th percentiles for the entire TQIP centers, are shown (adopted from the TQIP report).

**Table 1 healthcare-11-02865-t001:** TQIP patient cohort reporting definitions.

TQIP Cohorts	Definitions
Blunt Multisystem Injuries	∘ Blunt trauma type, derived from the submitted primary external cause code ∘ AIS severity ≥3 (3–6) in at least two of the following body regions: head, face, neck, thorax, abdomen, spine, upper or lower extremity
Penetrating Injuries	∘ Injury mechanism of cut/pierce or firearm, derived from the submitted external cause code ∘ Any injury with AIS severity ≥3 (3–6) in at least two of the following body regions: neck, thorax, abdomen
Hemorrhagic Shock	∘ Initial ED/hospital systolic blood pressure (SBP) between 0 and 90 mm Hg ∘ Received transfusion of blood within 4 h of admission
Severe Traumatic Brain Injury (TBI)	∘ Initial ED/hospital GCS total ≤ AIS severity ≥3 for a valid qualifying injury in the AIS head/body region ∘ Excludes isolated TBI AIS 05/08 codes listed in TQIP reporting code sets (patients are eligible for this cohort if they have another qualifying injury (i.e., if they have a brain injury and a code above, they may qualify for the cohort) of events ∘ No other injuries with an AIS severity of >2 in any other no-head AIS body region
Elderly Patients	∘ Age ≥65 years
Elderly Blunt Multisystem Injury	∘ Meets the cohort criteria for both elderly and blunt multisystem cohorts
Elderly Patients with Isolated Hip Fractures	∘ Age ≥65 years ∘ Injury mechanism of fall, derived from the submitted external cause code ∘ At least one of the AISo5/08 codes listed in TQIP reporting code sets ∘ Any other injuries are in AIS 05/08 codes listed in TQIP reporting code sets ∘ Any other injuries in AIS external body region (i.e., bruise, abrasion, or laceration)
Shock	∘ Initial ED/hospital SBP between 0 and 90 mm Hg
**TQIP Outcome Definitions**	
Major Hospital Events	At least one of the following 13 hospital events defined in the NTDS data dictionary: ∘ Acute Kidney Injury; ∘ Acute Respiratory Distress Syndrome; ∘ Cardiac Arrest with CPR; ∘ Central Line-Associated Blood Stream Infections (CLABSIs); ∘ Deep Surgical Site Infections; ∘ Myocardial Infractions; ∘ Organ Surgical Site Infections; ∘ Pressure Ulcer; ∘ Pulmonary Embolism; ∘ Severe Sepsis; ∘ Stroke/CVA; ∘ Unplanned Visit to the OR; ∘ Ventilator-Associated Pneumonia. Lived beyond 2 days after ED/hospital arrival
Mortality	One of the following discharge dispositions: ∘ ED discharge disposition of deceased/expired; ∘ Hospital discharge disposition of the deceased/expired; ∘ Hospital discharge disposition of discharged/transferred to hospice.
Major Hospital Events Including Death	Meet the outcome criteria for major hospital events/hospital arrival

All definitions were adopted from the ACS TQIP Benchmark Report References. FALL 2022 (Adult). Trauma Quality Improvement Program, September 2022.

**Table 2 healthcare-11-02865-t002:** Patients demographics, mechanisms, and modes of injury compared to the TQIP aggregate database.

	Fall 2020 (April 2019–March 2020)		Fall 2021 (April 2020–March 2021)	
Variables	All TQIP Centers	HTC TQIP Cohort	All TQIP Centers	HTC TQIP Cohort
Total No. of Patients; *n* (%)	325,102 (88.0%)	904 (98.8%)	349,080 (87.9%)	871 (98.5%)
Age (yrs.)	55 **	37 ± 13 *	53.7 ± 22.1	37.7 ± 14.1 *
Gender; *n* (%)				
Male	208,065 (64.0%)	830 (91.8) *	228,997 (65.6%)	818 (94.0%) *
Female	117,037 (36.0%)	74 (8.2) *	120,083 (34.4%)	53 (6.0%) *
Elderly (≥65 yrs)	122,732 (37.7%)	46 (5.0) *	123,076 (35.2%)	50 (5.74%) *
Race/Ethnicity; *n* (%)				
White	244,152 (75.1%)	289 (32.0) *	254,479 (72.9%)	274 (31.5%) *
Black	464,891 (14.3%)	63 (7) *	57,934 (16.6%)	71 (8.2%) *
Asian	7477 (2.3%)	551 (61.0) *	6981 (2.0%)	523 (60.0%) *
Others	23,732 (7.3%)	1 (0.1) *	26,181 (7.5%)	23 (0.3%) *
Unknown	11,053 (3.4%)	(0.0) *	12,567 (3.6%)	1 (0.1%) *
Mechanism of Injury; *n* (%)				
Blunt	46,997 (14.4%)	170 (18.8) *	52,136 (14.9%)	173 (19.9%) *
Penetrating	15,354 (4.7%)	20 (3.3) *	19,738 (5.7%)	22 (2.5%) *
Mode of Injury; *n* (%)				
Fall	151,172 (46.5%)	298 (33) *	155,340 (44.5%)	282 (32.4%) *
MVT Occupant and Other	71,847 (22.1%)	295 (32.6%) *	76,791 (22.0%)	261 (30.0%) *
MVT Motorcyclist	18,856 (5.8%)	37 (4.1%)	21,294 (6.1%)	67 (7.7%)
Pedestrian	23,732 (7.3%)	125 (13.8%) *	24,436 (7.0%)	94 (10.8%) *
Struck by/Against	15,605 (4.8%)	66 (7.3%) *	15,359 (4.4%)	85 (9.8%) *
Firearm	20,481 (6.3%)	4 (0.4%) *	28,973 (8.3%)	1 (0.1%) *
Cut/Pierce	8778 (2.7%)	24 (2.7%)	9774 (2.8%)	28 (3.2%)
Others	14,630 (4.5%)	55 (6.1%)	17,105 (4.9%)	52 (6.0%)

Data are expressed as count (percentage) or mean ± standard deviation whenever appropriate. * *p*-value < 0.05; ** standard deviation was not given in this report.

**Table 3 healthcare-11-02865-t003:** Comparison of comorbidities.

	Fall 2020		Fall 2021	
Variables	All TQIP Centers	HTC TQIP Cohort	All TQIP Centers	HTC TQIP Cohort
Pre-Existing Comorbidities	N (%)	N (%)	N (%)	N (%)
Hypertension	125,489 (38.6%)	96 (10.6%) *	129,159 (37.0%)	98 (11.3%) *
Smoking	74,123 (22.8%)	63 (7.0%) *	7086 (2.03%)	36 (4.13%) *
Dementia/Mental Disorder	53,317 (16.4%)	8 (0.9%) *	59,692 (17.1%)	13 (1.4%) *
Diabetes Mellitus	50,391 (15.5%)	98 (10.8%) *	52,013 (14.9%)	94 (10.8%) *
COPD	23,407 (7.2%)	0 (0.0%) *	23,737 (6.8%)	1 (0.1%) *
Alcohol Use Disorder	214,506 (6.6%)	6 (0.6%) *	82,034 (23.5%)	58 (6.7%) *
Congestive Heart Failure	14,304 (4.4%)	0 (0.0%) *	15,359 (4.4%)	1 (0.1%) *
Cerebrovascular Accident	9428 (2.9%)	3 (0.3%) *	9425 (2.7%)	3 (0.3%) *
Chronic Renal Failure	5852 (1.8%)	4 (0.4%) *	5934 (1.7%)	5 (0.5%) *
Cirrhosis	4551 (1.4%)	1 (0.1%) *	4887 (1.4%)	4 (0.5%) *
Myocardial Infraction	2276 (0.7%)	0 (0.0%) *	2095 (0.6%)	1 (0.1%) *
Disseminated Cancer	2094 (0.6%)	3 (0.3%)	2094 (0.6%)	3 (0.3%)
Acute Respiratory Distress syndrome	1626 (0.5%)	4 (0.4%)	1396 (0.4%)	4 (0.5%)

* *p*-value < 0.05.

**Table 4 healthcare-11-02865-t004:** Comparison of in-hospital complications.

	Fall 2020		Fall 2021	
Variables	All TQIP Centers	HTC TQIP Cohort	All TQIP Centers	HTC TQIP Cohort
Complications	N (%)	N (%)	N (%)	N (%)
Acute Kidney Injury	2563 (0.8%)	4 (0.4%)	3141 (0.9%)	6 (0.7%)
Pressure Ulcer	2242 (0.7%)	5 (0.5%)	2793 (0.8%)	3 (0.3%)
Deep vein Thrombosis	3524 (1.1%)	4 (0.4%)	3839 (1.1%)	2 (0.2%) *
Pulmonary Embolism	1602 (0.5%)	10 (1.1%) *	2094 (0.6%)	10 (1.1%) *
Catheter-Associated Urinary Tract Infection	961 (0.3%)	7 (0.8%) *	1047 (0.3%)	7 (0.8%) *
Ventilator-Associated Pneumonia	2563 (0.8%)	15 (1.7%) *	2793 (0.8%)	29 (3.3%) *
Severe Sepsis	1281 (0.4%)	2 (0.2%)	1745 (0.5%)	2 (0.2%)
Superficial Incisional Surgical Site Infection	641 (0.2%)	13 (1.4%) *	698 (0.2%)	10 (1.1%) *
Deep Surgical Site infection	641 (0.2%)	1 (0.1%)	698 (0.2%)	4 (0.5%)
Central Line Bloodstream Infection (CLABSI)	320 (0.1%)	5 (0.5%) *	349 (0.1%)	9 (1.0%) *

* *p*-value < 0.05.

**Table 5 healthcare-11-02865-t005:** Management of underlying shock; VTE parameters.

	Fall 2020		Fall 2021	
Variables	All TQIP Centers	HTC TQIP Cohort	All TQIP Centers	HTC TQIP Cohort
Hemorrhagic Shock	N (%)	N (%)	N (%)	N (%)
Number of Patients (%)	7766 (2.4%)	28 (3.1%)	9003 (2.6%)	35 (4.01%) *
PRBC Transfusion within 24 h	7700 (99.1%)	28 (100%)	8479 (94.1%)	35 (4.01%) *
Plasma Transfusion within 24 h	5571 (71.7%)	7 (25%) *	6184 (68.6%)	14 (40%)
Platelets Transfusion within 24 h	3322 (42.7%)	8 (28.6%)	3397 (37.7%)	12 (34.2%)
Surgery for Hemorrhagic Control	4066 (52.3%)	11 (39.2%)	4745 (52.8%)	19 (54.3%) *
Angiography for Hemorrhagic Shock	1327 (17.1%)	5 (17.8%)	1457 (16.2%)	10 (28.6%) *
Pharmacologic VTE Prophylaxis	217,794 (66.9%)	739 (81.7%) *	241,147 (71.8%)	734 (86.4%) *
Time to VTE Prophylaxis, Median (IQR)	2 (2–3)	2 (2–3)	2 (2–3)	2 (2–3)
Unfractionated Heparin	48,870 (20.1%)	3 (0.3%) *	47,005 (19.5%)	4 (0.4%) *
Low-Molecular-Weight Heparin	164,697 (75.6%)	732 (99.1%) *	184,626 (76.6%)	727 (99.0%) *

* *p*-value < 0.05.

**Table 6 healthcare-11-02865-t006:** Comparison of injury severity and outcome variables between HTC and TQIP aggregate database.

	Fall 2020		Fall 2021	
Variables	All TQIP Centers	HTC TQIP Cohort	All TQIP Centers	HTC TQIP Cohort
Injury Severity	N (%)	N (%)	N (%)	N (%)
Total GCS ≤ 8	37,062 (11.4%)	108 (12%)	40,144 (11.5%)	122 (14%) *
Injury Severity Score (ISS), Median (IQR)	14 (10–19)	14 (10–20)	14 (10–19)	14 (10–21)
Midline Shift TBI	17,454 (5%)	41 (4.7%)	17,454 (5%)	41 (4.7%)
Shock (SBP < 90 mm Hg)	13,869 (4.3%)	60 (6.6%) *	15,564 (3.9%)	56 (6.3%) *
Severe TBI (AIS ≥ 3 and GCS 3–8)	23,772 (7.3%)	87 (9.6%) *	26,480 (6.7%)	92 (10.4%) *
Pre-Hospital Cardiac Arrest	4226 (1.3%)	19 (2.1%) *	4887 (1.4%)	21 (2.4%) *
Outcome Variables				
Hospital Length of Stay, Median (IQR)	5 (3–9)	8 (5–14) *	5 (3–9)	8 (4–16) *
Patients with ICU Care (disposition)	159,295 (49.0%)	475 (52.5%) *	163,020 (46.7%)	486 (55.9%) *
ICU Length of Stay, Median (IQR)	3 (2–6)	5 (3–8) *	3 (2–6)	5 (3–9) *
Patients with Mechanical Ventilation	62,094 (19.1%)	190 (21%)	67,372 (19.3%)	202 (23.2%) *
Ventilatory Days, Median (IQR)	3 (2–8)	4 (2–9)	3 (2–8)	5 (2–11) *
Death within 72 h.	10,382 (43.3%)	0 (0%) *	166,511 (47.7%)	158 (18.2%) *
Death After 30 Days	623 (2.6%)	3 (8.3%) *	8378 (2.4%)	2 (0.2%) *
Time to Death, Median (IQR) Days	4 (2–9)	7 (4.5–10.5) *	4 (2–9)	7 (5–8) *
Overall Mortality	23,977 (7.4%)	36 (3.9%) *	27,577 (7.9%)	31 (3.6%) *

* *p*-value < 0.05.

**Table 7 healthcare-11-02865-t007:** Risk-adjusted mortality and major hospital events, including mortality for our cohort during the Fall 2020 and Fall 2021 periods.

TQIP Risk-Adjusted Mortality								
	Fall 2020				Fall 2021			
Cohort	Patients (*n*)	Observed Events; *n* (%)	TQIP Average	Odds Ratio (95% CI)	Patients (*n*)	Observed Events; *n* (%)	TQIP Average	Odds Ratio (95% CI)
All Patients	904	36 (4.0%)	7.4%	0.58 (0.40–0.84) *	871	31 (3.6%)	7.9%	0.47 (0.32–0.67) *
Blunt Multisystem	170	19 (11.2%)	15.1%	0.70 (0.45–1.09)	173	20 (11.6%)	14.9%	0.67 (0.44–1.03)
Penetrating	20	1 (5.0%)	7.8%	0.89 (0.36–2.19)	22	00 (0.0%)	10.8%	0.75 (0.31–1.85)
Shock	60	15 (25.0%)	26.8%	0.89 (0.58–1.36)	56	12 (21.4%)	27.4%	0.78 (0.51–1.20)
Severe TBI	87	28 (32.2%)	45.5%	0.64 (0.40–1.01)	92	18 (19.6%)	46.0%	0.42 (0.26–0.68) *
Elderly	46	6 (13.0%)	10.1%	1.10 (0.63–1.92)	50	6 (12.0%)	10.9%	1.01 (0.61–1.67)
Elderly Blunt Multisystem	5	1 (20.0%)	21.9%	0.99 (0.69–1.43)	6	2 (33.3%)	21.9%	1.06 (0.63–1.77)
Isolated Hip Fracture	11	0 (0.0%)	3.2%	0.98 (0.47–2.03)	13	0 (0.0%)	3.8%	0.97 (0.49–1.92)
**TQIP Risk-Adjusted Major Hospital Events**								
**Cohort**	**Fall 2020**				**Fall 2021**			
All Patients	876	66 (7.5%)	11.1%	0.80 (0.59–1.07)	871	79 (9.1%)	11.8%	0.77 (0.59–1.02)
Blunt Multisystem	160	33 (20.6%)	24.5%	0.88 (0.59–1.32)	173	43 (24.9%)	24.4%	0.94 (0.65–1.36)
Penetrating	18	2 (11.1%)	19.4%	0.97 (0.52–1.80)	22	3 (13.6%)	20.0%	1.09 (0.55–2.16)
Shock	56	24 (42.9%)	37.9%	1.04 (0.69–1.58)	56	18 (32.1%)	38.8%	0.77 (0.51–1.17)
Severe TBI	82	41 (50.0%)	55.1%	0.88 (0.58–1.36)	92	38 (41.3%)	56.0%	0.72 (0.46–1.11)
Elderly	44	10 (22.7%)	12.7%	1.46 (0.87–2.46)	50	7 (14.0%)	13.6%	0.96 (0.61–1.52)
Elderly Blunt Multisystem	4	2 (50.0%)	28.4%	1.03 (0.77–1.37)	6	3 (50.0%)	28%	1.07 (0.69–1.66)
Isolated Hip Fracture	11	0 (0%)	5%	0.96 (0.49–1.89)	13	3 (23.1%)	5.3%	1.36 (0.69–2.70)

TBI: Traumatic Brain Injury; * significant odds ratio.

## Data Availability

All data are presented in the manuscript and tables.

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
