# Peer review of "Trauma Quality Improvement Program: A Retrospective Analysis from A Middle Eastern National Trauma Center"

_healthcare, 2023, doi:10.3390/healthcare11212865_

Round 1
Reviewer 1 Report
This paper provides a good example of how data gathered within healthcare registries can provide important information to drive quality improvement. Overall, it is clear, well structured, presenting an adequate description of the study and its results, but there are a few aspects that could be improved:
Provide a clearer explanation of the aims of the paper. In the abstract the authors state that they aimed “to analyze the clinical outcomes of TQIP benchmarks in trauma care at the Level I Hamad trauma center (HTC) compared with the aggregate TQIP (all 19 centers) data”, but they do not clearly state the purpose of this analysis. Why it is important, and what do the authors expected to find?
Readers who are less familiar with trauma care would benefit from a brief explanation of the differences between level I and II trauma centres (p. 2, ln. 61). Also, that sentence should read “TQIP achieves its objectives by collecting data from level I and II trauma centres”.
Using the term “Fall” to describe 12-month-data collection periods happening between April and March may be confusing to some international audiences. Perhaps simply referring to the year in question would be better (i. e. 2020; 2021), or Year 1 – Year 2.
It would be beneficial if the authors could provide a little bit of information about the characteristics of a national tertiary trauma centre in Qatar. For example, what does this mean in terms of patients’ characteristics, and accessibility to care and treatment within that facility (e. g. is it primarily driven by need / severity, availability of funding, regional location, other…)?
There is considerable overlap between the information provided in the manuscript and the information provided on many of the tables. For example, the reporting definitions shown on Table 1 largely overlap with the inclusion criteria, and some results are reported both in the body of the manuscript and on tables (e. g. Table 2 vs. ln. 185-199). Please revise the manuscript to avoid repetition, and instead explicitly refer to the tables where the detailed information is provided.
Ensure all acronyms are defined when mentioned for the first time throughout the text (e. g. p. 1. ACS, p. 5. OR, p. 8. PRBC, etc.).
Within the statistical analysis section briefly describe which statistical estimates / methods are used when / why (e. g. types of data, nature of the distribution).
Explain why, with so many comparisons, the P value was set at .05.
Briefly describe what calculating the estimates of risk-adjusted mortality entails.
In the discussion (ln. 348-349), the authors state that there should be continued efforts to decrease ventilatory days, length in ICU and overall hospital stay. However, have they considered the possibility that these increased stays may be one of the factors contributing to the better performance observed in their hospital?
Minor revisions:
p. 3, ln. 104 – Replace “cohort” with “cohorts” at the end of the sentence.
p. 3, ln. 105-106 – This sentence is unclear. It should either read “a summary of TQIP cohorts” OR “a summary of the TQIP cohorts patient’s characteristics… is outlined…”
p. 6. ln. 183 – provide citation for the EpiInfo Software.
p. 6, ln. 198 – Replace “as” with “and” before “blunt trauma was significantly higher”.
p. 6, Table 2 – To facilitate reading and avoid confusion, change the symbol for the missing standard deviations to one that is more distinctive from the asterisk used to identify significant results.
p. 7, Table 3 – Consider sorting the variables on the table in alphabetical order, or from most to least prevalent within All TQIP Centres.
p. 8, ln. 219 – Edit the sentence to read “with up to half dying prior to arrival to…”
p. 8., ln. 226 – Edit the sentence to read “within 24 hrs than those in the TQIP database (Table 5).”
p. 8, ln. 227 – Replace “receiving” with “received”.
p. 9, ln. 233 – Delete “a” before “hospital stays” such that the sentence reads “and hospital stays longer than…”
p. 9, Table 6 – Indicate the unit for the Time to Death variable (is it days?).
p. 10, Table 7 – Flag significant Odds Ratios. Use the same labelling as on previous tables (e. g. explicitly indicate which columns refer to the HTC cohorts).
p. 11, Figure 2 – Increase the font size on all panels of the figure. On lines 259 and 264 of the caption, replace “had” with “was”. Revise the last sentence of the caption, as its unclear.
p. 11, ln. 275 – There is reference to “real-time retrospective data”. These two properties seem contradictory. Please clarify.
p. 13, Action plan – Use tense consistently (past to describe the findings / what the data revealed, or action plans that had been implemented at the time of writing; future tense to describe action plans that have been developed, but yet to be implemented – e. g. These areas were / will be prioritised and scrutinized to rule out…
p. 13, ln. 379 – This sentence needs revision. Do the authors mean “shared” rather than “shredded”? Also, the verb “educated” does not really work when the subject is “agreed-upon changes”.
p. 13, ln. 387 – Delete “small”, and further edit the sentence such that it reads “The sample size from HTC was another limitation”. Ns of 871 to 904 are not exactly small. The point that some sub-groups within the overall sample may be small is clearly made on the remainder of the paragraph (ln. 388-392).
By and large the paper is well written, with only minor edits of the English language required. Please check the use of articles or lack thereoff (e. g. p. 9, ln. 235), verb tenses, and "turns of phrase" (e. g. p. 9, ln. 237 should read "was found to be like other TQIP", rather than "to be like to other") throughout the paper.
Author Response
This paper provides a good example of how data gathered within healthcare registries can provide important information to drive quality improvement. Overall, it is clear, well structured, presenting an adequate description of the study and its results, but there are a few aspects that could be improved:
We would like to thank the reviewers for their valuable inputs and feedback to improve our manuscript further. We have responded to the queries of the reviewers in point wise fashion and the related corrections has been incorporated in the manuscript appropriately with track changes.
Thank you.
Query: Provide a clearer explanation of the aims of the paper. In the abstract the authors state that they aimed “to analyze the clinical outcomes of TQIP benchmarks in trauma care at the Level I Hamad trauma center (HTC) compared with the aggregate TQIP (all 19 centers) data”, but they do not clearly state the purpose of this analysis. Why it is important, and what do the authors expected to find?
Response: We have refined the paper's objective to provide a more precise purpose and outlined our anticipated outcomes for this manuscript.
Query: Readers who are less familiar with trauma care would benefit from a brief explanation of the differences between level I and II trauma centres (p. 2, ln. 61). Also, that sentence should read “TQIP achieves its objectives by collecting data from level I and II trauma centres”.
Response: We agree with the reviewer, and have modified the sentence. Moreover, we have briefly explained and added the difference between level1 and II trauma centers in the introduction section.
Query: Using the term “Fall” to describe 12-month-data collection periods happening between April and March may be confusing to some international audiences. Perhaps simply referring to the year in question would be better (i. e. 2020; 2021), or Year 1 – Year 2.
Response: Done
Query: It would be beneficial if the authors could provide a little bit of information about the characteristics of a national tertiary trauma centre in Qatar. For example, what does this mean in terms of patients’ characteristics, and accessibility to care and treatment within that facility (e. g. is it primarily driven by need / severity, availability of funding, regional location, other…)?
Response: We have briefly explained about the Hamad trauma center (HTC) in materials and methods section.
Query: There is considerable overlap between the information provided in the manuscript and the information provided on many of the tables. For example, the reporting definitions shown on Table 1 largely overlap with the inclusion criteria, and some results are reported both in the body of the manuscript and on tables (e.g. Table 2 vs. ln. 185-199). Please revise the manuscript to avoid repetition, and instead explicitly refer to the tables where the detailed information is provided.
Response: Table 1 is the summary of TQIP patient cohort standard definitions, which TQIP uses. The inclusion criteria is for blunt and penetrating injuries and we have modified the inclusion criteria. Some of the text-explaining table 2 were removed to avoid repetitions.
Query: Ensure all acronyms are defined when mentioned for the first time throughout the text (e. g. p. 1. ACS, p. 5. OR, p. 8. PRBC, etc.).
Response: Done
Query: Within the statistical analysis section briefly describe which statistical estimates / methods are used when / why (e.g. types of data, nature of the distribution).Explain why, with so many comparisons, the P value was set at .05. Briefly describe what calculating the estimates of risk-adjusted mortality entails.
Response: We have already mentioned the statistical methods with type of data used in the statistical analysis section in detail. We took p value < 0.05 as it best balances the risk of making type 1 and type 2 errors.
Assessing the performance of trauma care necessitates the use of validated risk-adjustment techniques. Risk-adjusted mortality following trauma involves evaluating the observed mortality rate in comparison to the expected mortality rate, which is determined by a statistical model incorporating various factors associated with trauma-related fatalities.
The purpose of risk adjustment is to eliminate sources of variability that are not linked to the institution's practices, aiming to ensure that any remaining disparities truly represent variations in the quality of care provided.
We have modified the statistical section and have included the risk-adjusted mortality analysis statement.
Query: In the discussion (ln. 348-349), the authors state that there should be continued efforts to decrease ventilatory days, length in ICU and overall hospital stay. However, have they considered the possibility that these increased stays may be one of the factors contributing to the better performance observed in their hospital?
Response: While we cannot dismiss this possibility entirely, we would like to emphasize a few important points .Yes, we have considered that increased ventilator days and ICU/hospital stays may be ‘the cost’ of attaining better outcomes when compared to the other TQIP centers. However, attributing these gains to longer ICU and hospital stays and more ventilator days is difficult when quality improvement programs focus on processes that are proven to improve outcomes, e.g. reducing ventilatory days, length in ICU and overall hospital stay. We are more of the mind that these process indicators reflect the level of care that our sicker patient population needs. Our patient population has significantly more patients with a GCS <8, SBP < 90mmHg, severe TBI and pre-hospital cardiac arrest; yet our patients have significantly lower overall mortality and early death [within 72 hours] rates. These patients simply need more time to receive the necessary quality and quantity of care, in order to attain better outcomes.
Furthermore, we have not presented a comparison of the quality improvements that occurred before and after our institution was included in the ACS-TQIP database, which would have offered a more accurate assessment of our center's performance in terms of outcome measures. The data presented in our manuscript focuses solely on the improvements and concerns identified in the Fall-2021 versus Fall-2020 TQIP report. However, it important to note that TQIP reports are issued biannually, and it may take several reporting cycles to observe and comment on factors responsible for improvements in TQIP metrics following the implementation of corrective actions. For this, we have to compare our performance with TQIP aggregate for more than one cycle.
The key is to take things one-step at a time. We expect that consistent progress will ultimately lead to substantial enhancements in quality, outcomes, and patient safety at our trauma center.
Minor revisions:
Query: p. 3, ln. 104 – Replace “cohort” with “cohorts” at the end of the sentence.
Response: Done
Query: p. 3, ln. 105-106 – This sentence is unclear. It should either read “a summary of TQIP cohorts” OR “a summary of the TQIP cohorts patient’s characteristics… is outlined…”
Response: Done
Query : p. 6. ln. 183 – provide citation for the EpiInfo Software.
Response: Done
Query : p. 6, ln. 198 – Replace “as” with “and” before “blunt trauma was significantly higher”.
Response: The text has been removed as per the previous request to eliminate redundant information in Table 2.
Query : p. 6, Table 2 – To facilitate reading and avoid confusion, change the symbol for the missing standard deviations to one that is more distinctive from the asterisk used to identify significant results.
Response: Done
Query : p. 7, Table 3 – Consider sorting the variables on the table in alphabetical order, or from most to least prevalent within All TQIP Centres.
Response: Done. We have sorted the variables based on the prevalence.
Query : p. 8, ln. 219 – Edit the sentence to read “with up to half dying prior to arrival to…”
Response: Done
Query : p. 8., ln. 226 – Edit the sentence to read “within 24 hrs than those in the TQIP database (Table 5).”
Response: Done
Query : p. 8, ln. 227 – Replace “receiving” with “received”.
Response: Done
Query : p. 9, ln. 233 – Delete “a” before “hospital stays” such that the sentence reads “and hospital stays longer than…”
Response: Done
Query : p. 9, Table 6 – Indicate the unit for the Time to Death variable (is it days?).
Response: Done
Query : p. 10, Table 7 – Flag significant Odds Ratios. Use the same labelling as on previous tables (e. g. explicitly indicate which columns refer to the HTC cohorts).
Response: Done
Query: p. 11, Figure 2 – Increase the font size on all panels of the figure. On lines 259 and 264 of the caption, replace “had” with “was”. Revise the last sentence of the caption, as its unclear.
Response: Done
Query: p. 11, ln. 275 – There is reference to “real-time retrospective data”. These two properties seem contradictory. Please clarify.
Response: It may seem contradictory, but TQIP employs both real-time monitoring and retrospective data analysis for tracking trends and improvements, which is essential. Typically, we collect retrospective data to assess improvements in quality and patient safety, such as tracking infection rates and patient falls. This historical data informs us about past performance. On the other hand, we also gather real-time monitoring data to address current needs and plan for future situations, helping us identify and mitigate risks in real time. TQIP's approach maximizes patient safety by effectively utilizing both types of data concurrently.
Query : p. 13, Action plan – Use tense consistently (past to describe the findings / what the data revealed, or action plans that had been implemented at the time of writing; future tense to describe action plans that have been developed, but yet to be implemented – e. g. These areas were / will be prioritised and scrutinized to rule out…
Response: Done
Query : p. 13, ln. 379 – This sentence needs revision. Do the authors mean “shared” rather than “shredded”? Also, the verb “educated” does not really work when the subject is “agreed-upon changes”.
Response: We have replaced the word“shredded” with “scrutinized”.
Query p. 13, ln. 387 – Delete “small”, and further edit the sentence such that it reads “The sample size from HTC was another limitation”. Ns of 871 to 904 are not exactly small. The point that some sub-groups within the overall sample may be small is clearly made on the remainder of the paragraph (ln. 388-392).
Response: Done
Query: Comments on the Quality of English Language
By and large the paper is well written, with only minor edits of the English language required. Please check the use of articles or lack there off (e. g. p. 9, ln. 235), verb tenses, and "turns of phrase" (e. g. p. 9, ln. 237 should read "was found to be like other TQIP", rather than "to be like to other") throughout the paper
Response: We have revisited the manuscript and have edited the manuscript as per reviewers suggestion
Reviewer 2 Report
Trauma Quality Improvement Program (TQIP): Insights from A Middle Eastern National Trauma Center
Thank you very much for allowing me to review the manuscript. Below, I offer some suggestions for improvement in preparation for publication.
Title: Specify the study design in the title.
Methods:
• Please present the inclusion and exclusion criteria section in prose rather than using bullet points.
• Table 1 is not clear. Perhaps justifying the text on both sides would enhance clarity.
• The analyses were not disaggregated by gender. This is necessary and has been internationally demanded in recent decades. Please consider conducting separate analyses or provide justification if this has been done previously and no differences were found between them.
Discussion:
• Specify the strengths and implications for clinical practice arising from the results.
Author Response
Query: Title: Specify the study design in the title.
Response: Done
Methods: Query: Please present the inclusion and exclusion criteria section in prose rather than using bullet points.
Response: Done
- Table 1 is not clear. Perhaps justifying the text on both sides would enhance clarity.
Response: Done
Query: The analyses were not disaggregated by gender. This is necessary and has been internationally demanded in recent decades. Please consider conducting separate analyses or provide justification if this has been done previously and no differences were found between them.
Response: We acknowledge the reviewer's point regarding the potential influence of gender disparities on the social and economic consequences of injury-related disabilities. However, it's important to note that in many Gulf countries, social and cultural norms have resulted in a significant overrepresentation of males in injury cases, masking the trauma's impact on the female population. This trend is consistent with our current analysis, where females constituted only a small percentage (6-8%) of the total trauma cohort. It's worth mentioning that our current study did not specifically focus on gender-based analysis as it was beyond our scope. Nevertheless, future research should prioritize investigating the primary causes of injury, variations in treatment, complications, and long-term outcomes among females in more depth.
Query: Discussion:
- Specify the strengths and implications for clinical practice arising from the results.
Response: Strength and clinical Implications: In the TQIP Fall 2020 report, HTC was considered a strong performer (low outlier) in one cohort (all patients) and performed at an average level in the remaining cohorts. However, in the Fall 2021 report, we observed significant progress, with HTC being categorized as a strong performer (low outlier) in two cohorts (all patients and severe TBI patients) for overall mortality. This improvement over time in risk-adjusted mortality underscores the dedicated and continuous efforts of our trauma team. Analyzing and comparing our data with the ACS TQIP database has enabled us to identify and assess trends and outliers. This approach allows us to track these trends over time and pinpoint any outlier instances, subsequently enabling the implementation of corrective action plans. After these corrective actions are put into place, reevaluating the latest TQIP report would informs us whether the initially identified issues have improved. This analysis is instrumental in determining the impact of our action items and whether a shift in approach is necessary.
Round 2
Reviewer 2 Report
Thank you very much for allowing me to review this manuscript again. The authors have done a good job in addressing the requested comments. They have resolved all my concerns.